# Multimorbidity, polypharmacy, and COVID-19 infection within the UK Biobank cohort

Ross McQueenie[1‡], Hamish M. E. Foster[1‡], Bhautesh D. Jani[1], Srinivasa Vittal Katikireddi[1], Naveed Sattar[2], Jill P. Pell[1], Frederick K. Ho[1], Claire L. Niedzwiedz[1], Claire E. Hastie[1], Jana Anderson[1], Patrick B. Mark[2], Michael Sullivan[2], Catherine A. O'Donnell[1], Frances S. Mair[1‡], Barbara I. Nicholl[1‡]*

**1** Institute of Health and Wellbeing, College of Medical, Veterinary and Life Sciences, University of Glasgow, Glasgow, United Kingdom, **2** British Heart Foundation Glasgow Cardiovascular Research Centre, Institute of Cardiovascular and Medical Sciences, University of Glasgow, Glasgow, United Kingdom

‡ RM and HMEF share first authorship on this work. FSM and BIN are joint senior authors on this work.
* barbara.nicholl@glasgow.ac.uk

**Data Availability Statement:** The data in this study is owned by the UK Biobank (www.ukbiobank.ac.uk) and as researchers we are not entitled to republish or otherwise make available any UK

## Abstract

### Background

It is now well recognised that the risk of severe COVID-19 increases with some long-term conditions (LTCs). However, prior research primarily focuses on individual LTCs and there is a lack of data on the influence of multimorbidity ($\geq$2 LTCs) on the risk of COVID-19. Given the high prevalence of multimorbidity, more detailed understanding of the associations with multimorbidity and COVID-19 would improve risk stratification and help protect those most vulnerable to severe COVID-19. Here we examine the relationships between multimorbidity, polypharmacy (a proxy of multimorbidity), and COVID-19; and how these differ by sociodemographic, lifestyle, and physiological prognostic factors.

### Methods and findings

We studied data from UK Biobank (428,199 participants; aged 37–73; recruited 2006–2010) on self-reported LTCs, medications, sociodemographic, lifestyle, and physiological measures which were linked to COVID-19 test data. Poisson regression models examined risk of COVID-19 by multimorbidity/polypharmacy and effect modification by COVID-19 prognostic factors (age/sex/ethnicity/socioeconomic status/smoking/physical activity/BMI/systolic blood pressure/renal function). 4,498 (1.05%) participants were tested; 1,324 (0.31%) tested positive for COVID-19. Compared with no LTCs, relative risk (RR) of COVID-19 in those with 1 LTC was no higher (RR 1.12 (CI 0.96–1.30)), whereas those with $\geq$2 LTCs had 48% higher risk; RR 1.48 (1.28–1.71). Compared with no cardiometabolic LTCs, having 1 and $\geq$2 cardiometabolic LTCs had a higher risk of COVID-19; RR 1.28 (1.12–1.46) and 1.77 (1.46–2.15), respectively. Polypharmacy was associated with a dose response higher risk of COVID-19. All prognostic factors were associated with a higher risk of COVID-19 infection in multimorbidity; being non-white, most socioeconomically deprived, BMI $\geq$40 kg/m2, and reduced renal function were associated with the highest risk of COVID-19 infection: RR 2.81 (2.09–3.78); 2.79 (2.00–3.90); 2.66 (1.88–3.76); 2.13 (1.46–3.12), respectively. No

Biobank data at the individual participant level (https://www.ukbiobank.ac.uk/wpcontent/uploads/2013/10/ukbiobank-data-management.pdf). However, any bona fide researcher can apply to use the UK Biobank resource for health-related research that is in the public interest by following the registration and access link https://bbams.ndph.ox.ac.uk/ams/.

**Funding:** The author(s) received no specific funding for this work.

**Competing interests:** JPP is a member of the Scientific Advisory Committee of UK Biobank. This does not alter our adherence to PLOS ONE policies on sharing data and materials. All other authors have no potential, perceived, or real conflicts of interest.

multiplicative interaction between multimorbidity and prognostic factors was identified. Important limitations include the low proportion of UK Biobank participants with COVID-19 test data (1.05%) and UK Biobank participants being more affluent, healthier and less ethnically diverse than the general population.

## Conclusions

Increasing multimorbidity, especially cardiometabolic multimorbidity, and polypharmacy are associated with a higher risk of developing COVID-19. Those with multimorbidity and additional factors, such as non-white ethnicity, are at heightened risk of COVID-19.

## Introduction

COVID-19 is an ongoing pandemic caused by the severe acute respiratory syndrome coronavirus 2 (SARS-CoV-2) [1–3]. COVID-19 clinical manifestations range from asymptomatic and mild upper respiratory symptoms to severe respiratory failure and death [4]. A range of prognostic factors for greater mortality from COVID-19 have been identified including age [5], male sex [6], non-white ethnicity [7], obesity [8], pre-existing long-term conditions (LTCs; e.g. hypertension [9], diabetes [10], chronic kidney disease [11]), and multimorbidity (presence of ≥2 LTCs) [11–13]. As a result, a key mitigation strategy in many countries is the identification and protection of those deemed vulnerable or at higher risk of adverse outcomes [14].

For example, in both the UK and the USA, age (≥70 in UK and ≥65 in USA), severe obesity (BMI ≥40 kg/m$^2$), being immunocompromised, and certain LTCs (e.g. cardiac/respiratory/renal disease) are used to identify those at higher risk of severe disease [14, 15]. Individuals who meet any of these criteria have been asked to be more cautious and adhere to public health guidance (e.g. social distancing) more strictly than the general population. In the UK, this higher-risk group is distinct from those with specific LTCs (e.g. leukaemia) who are considered 'extremely high risk' and, as a result, are asked to 'shield' and not leave their homes at all [16].

To date, LTC prognostic factors for severe COVID-19 primarily involve single conditions and there is a lack of data on the influence of multimorbidity on the risk of COVID-19 [5, 13, 17]. Multimorbidity prevalence is increasing worldwide and is associated with higher mortality, with different disease clusters associated with even higher mortality [18–21]. Polypharmacy, closely linked to multimorbidity [22], is also associated with adverse health outcomes [23]. Therefore, it is plausible that the number of LTCs, type of LTCs, and polypharmacy are associated with a higher risk of developing severe COVID-19. Furthermore, it is plausible that the adverse consequences of multimorbidity on COVID-19 risk may be greater for population subgroups (e.g. people aged ≥65 years, or those with a BMI ≥40kg/m$^2$). Such knowledge would benefit clinicians, as this is routinely collected information in many clinical settings. Using a large UK population cohort, UK Biobank, we aimed to investigate:

1. the association between multimorbidity (by number and type of LTC, and by level of polypharmacy) and COVID-19.

2. the potential effect modification of known COVID-19 sociodemographic and physiological risk factors on the association between multimorbidity and COVID-19.

## Methods

### Study design and data collection

Data came from UK Biobank, a longitudinal population-based cohort of 502,503 participants aged between 37–73 years old at baseline from England, Wales and Scotland [24]. Baseline data were collected across 22 assessment centres between 2006–2010. UK Biobank contains detailed biological measurements and self-reported demographic, lifestyle, and health information elicited by touch-screen questionnaire and nurse-led interview. COVID-19 test samples were collected and processed between 16 March 2020 and 18[th] May 2020. COVID-19 test results were provided by Public Health England (PHE) [25]. Data presented here are from participants recruited from 16 assessment centres located in England only (5 participants with COVID-19 test data were excluded as they attended baseline assessment centres in Scotland or Wales). Participants who had died prior to the last available mortality register extraction (14 February 2018) were also excluded. This resulted in a final eligible study population of 428,199 participants.

This study was conducted as part of UK Biobank project number 14151 and is covered by the generic ethics approval for UK Biobank studies from the NHS National Research Ethics Service (16/NW/0274).

### Outcomes

Our outcome of interest was confirmed COVID-19 infection (defined as at least one positive test result). Whether or not participants received a COVID-19 test was used as an outcome for a secondary analysis, presented in Supplementary Material (S1–S3 Tables).

### Measurement of LTCs

Physician-diagnosed LTCs were self-reported and confirmed at nurse-led interview at baseline. Number of LTCs, based on 43 commonly occurring LTCs from previous UK Biobank work [21], were categorised into 0, 1, and ≥2. When used as a predictor variable for COVID-19, LTCs were modelled using three measures: total number of LTCs; number of cardiometabolic LTCs (diabetes, coronary heart disease, atrial fibrillation, chronic heart failure, chronic kidney disease, hypertension, stroke/transient ischaemic attack, or peripheral vascular disease); and number of respiratory LTCs (asthma, chronic obstructive pulmonary disease, chronic bronchitis, emphysema, or bronchiectasis).

### Measurement of polypharmacy

Participant medication numbers were based on self-report at baseline and categorised into: 0, 1–3, 4–6, 7–9 and, ≥10 medications.

### Exposures

As for LTC and polypharmacy measures, all exposures were based on assessment at the time of recruitment. Sex was collected as a binary variable (male/female). Age at time of COVID-19 test was calculated using age at baseline, and dates of baseline assessment and COVID-19 test extract, and categorised as 48–59, 60–69, and 70–86 years. Ethnicity was self-reported and categorised as Asian/Asian British, black/black British, Chinese, mixed, white and other. Socioeconomic deprivation was measured using the Townsend score of participants' postcode of residence derived from Census data on car ownership, household occupancy, unemployment, and occupation and categorised into quintiles [26]. Smoking status was dichotomised as never and current/former. Frequency of alcohol intake was categorised into: "Never or special

occasions only", "1–3 times a month", "1–4 times a week", or "Daily or almost daily". Level of physical activity was defined as "none", "low", "medium", or "high" using Metabolic Equivalent Task (MET) scores based on the International Physical Activity Questionnaire (IPAQ) scoring protocol [27]. Assessment centre location, a categorical variable, describes attendance at one of sixteen assessment centres included in this study. BMI was derived from weight and height and categorised as <18.5, 18.5–25, 25–30, 30–35, or >35 kg/m$^2$. The UK Biobank study protocol is described in more detailed online [28].

To assess potential effect modification on the association between multimorbidity and COVID-19, known COVID-19 risk factors were re-categorised into dichotomous variables based on at-risk status: age (</≥65 years), ethnicity (white/non-white), physical activity (</≥ current UK guidelines of 150 min/week moderate or 75 min/week vigorous physical activity) [29], and BMI (</≥40kg/m$^2$; severely obese) [15]. Due to previously identified associations with severe COVID-19, two additional physiological risk factors were also included in this analysis: raised systolic blood pressure (</≥140 mmHg) as a marker of hypertension [9]; and reduced estimated glomerular filtration rate (eGFR; </≥60 ml/min/1.73 m$^2$), with <60 ml/min/1.73 m$^2$ used as a marker of chronic kidney disease [30]. eGFR was calculated by the CKD-EPI equation [31].

## Statistical analyses

We compared participants who tested positive for COVID-19 with those who tested negative or were not tested, based on sociodemographics (age, sex, Townsend score and ethnicity), lifestyle (smoking status, frequency of alcohol intake, physical activity), BMI, LTCs, and medication counts using $\chi^2$ tests. Poisson regression models were then used to test for an association between outcome measure (confirmed COVID-19 infection vs. no COVID-19 infection, including those with a negative result and those not tested) and number of LTCs (all, cardiometabolic and respiratory LTCs)/level of polypharmacy. We chose Poisson regression models rather than logistic regression to provide more interpretable relative risks as opposed to odds ratios [32]. Participants with no LTCs, no cardiometabolic LTCs, no respiratory LTCs, or not taking any medication formed the respective reference groups. Two models were conducted, adjusting for 1) sociodemographic variables (as above plus assessment centre location), and 2) as for model 1 adjustments plus lifestyle variables and BMI. For both $\chi^2$ tests and Poisson regression models, p<0.01 was considered statistically significant.

Next, we assessed whether there was evidence of effect modification on an additive scale by examining how the association between multimorbidity and COVID-19 differed across strata of known COVID-19 risk factors: sex, age, ethnicity, Townsend score, smoking, physical activity, BMI, systolic blood pressure, and eGFR. For each risk factor, the reference category was participants in the lowest risk category for that risk factor who also had no LTCs. We used Poisson regression models to examine the association between LTC and COVID-19 across the other combinations of LTC and risk factor categories. We tested formally for interactions by performing ANOVA tests between two models for each risk factor: one model containing an interaction term between the risk factor and number of LTCs, and one without the interaction term. Interactions were considered significant if p<0.01 for each ANOVA. As a secondary analysis we re-ran the analyses with the outcome as tested vs. not tested for COVID-19. The number of participants in each model varied depending on the proportion missing data for any included variable, however, the maximum proportion of missing data for any variable of interest was 2.9% (N = 12,350) for systolic blood pressure. All analyses were conducted using R studio v.1.2.1335 operating R v.3.6.1.

# Results

## Demographic and lifestyle factors

Of 428,199 eligible participants, 1,324 (0.31%) tested positive for COVID-19. Participants who tested positive for COVID-19 were older and more likely to be male, non-white, in the most deprived quintile, current/former smokers, drink alcohol rarely, to have a BMI of $\geq 40$ kg/m$^2$, do no physical activity, to have more LTCs (including cardiometabolic and respiratory LTCs), and to be taking more medications, compared to those who did not have a positive COVID-19 test (Table 1).

**Table 1. Cohort characteristics by COVID-19 test positive or not.**

| | COVID-19 test negative or not tested | COVID-19 test positive |
|---|---|---|
| | (n = 426,875) | (n = 1,324) |
| **Sex** | | |
| Female | 234,507 | 628 |
| | 54.9 % | 47.4 % |
| Male | 192,368 | 696 |
| | 45.1 % | 52.6 % |
| **Age at COVID-19 test (years)** | | |
| 48–59 | 94,652 | 381 |
| | 22.2 % | 28.8 % |
| 60–69 | 139,817 | 307 |
| | 32.8 % | 23.2 % |
| 70–86 | 192,406 | 636 |
| | 45.1 % | 48.0 % |
| **Ethnicity** | | |
| White | 399,388 | 1,139 |
| | 94.1 % | 86.7 % |
| Asian or Asian British | 9,186 | 60 |
| | 2.2 % | 4.6 % |
| Black or Black British | 7,650 | 76 |
| | 1.8 % | 5.8 % |
| Chinese | 1,396 | 6 |
| | 0.3 % | 0.5 % |
| Mixed | 2,652 | 9 |
| | 0.6 % | 0.7 % |
| Other ethnic group | 4,181 | 23 |
| | 1.0 % | 1.8 % |
| **Townsend quintile** | | |
| 1 (least deprived) | 84,840 | 179 |
| | 19.9 % | 13.5 % |
| 2 | 86,510 | 207 |
| | 20.3 % | 15.6 % |
| 3 | 85,788 | 222 |
| | 20.1 % | 16.8 % |
| 4 | 85,402 | 290 |
| | 20.0 % | 21.9 % |
| 5 (most deprived) | 83,835 | 425 |
| | 19.7 % | 32.1 % |

(*Continued*)

**Table 1.** (Continued)

| | COVID-19 test negative or not tested (n = 426,875) | COVID-19 test positive (n = 1,324) |
|---|---|---|
| **Smoking status** | | |
| Never | 235,056 | 642 |
| | 55.4 % | 49 % |
| Current or Previous | 189,299 | 669 |
| | 44.6 % | 51.0 % |
| **Frequency of alcohol intake** | | |
| Never or special occasions only | 82,785 | 363 |
| | 19.5 % | 27.5 % |
| One to three times a month | 47,535 | 183 |
| | 11.2 % | 13.9 % |
| One to four times a week | 208,046 | 563 |
| | 48.9 % | 42.7 % |
| Daily or almost daily | 87,211 | 209 |
| | 20.5 % | 15.9 % |
| **Physical activity level** | | |
| None | 25,887 | 157 |
| | 6.2 % | 12.2 % |
| Low | 15,687 | 51 |
| | 3.7 % | 4.0 % |
| Medium | 335,775 | 957 |
| | 79.8 % | 74.5 % |
| High | 43,383 | 120 |
| | 10.3 % | 9.3 % |
| **BMI (kg/m$^2$)** | | |
| <18.5 | 2,138 | 7 |
| | 0.5 % | 0.5 % |
| 18.5–25 | 135,563 | 297 |
| | 31.9 % | 22.7 % |
| 25–30 | 182,243 | 551 |
| | 42.9 % | 42.1 % |
| 30–35 | 75,201 | 282 |
| | 17.7 % | 21.5 % |
| ≥35 | 29,210 | 172 |
| | 6.9 % | 13.1 % |
| **Number of long-term conditions** | | |
| 0 | 148,826 | 351 |
| | 35.0 % | 26.8 % |
| 1 | 139,963 | 385 |
| | 32.9 % | 29.4 % |
| ≥ 2 | 136,508 | 572 |
| | 32.1 % | 43.7 % |
| **Number of cardiometabolic long-term conditions** | | |
| 0 | 300,363 | 773 |
| | 70.4 % | 58.4 % |
| 1 | 103,185 | 394 |
| | 24.2 % | 29.8 % |

(*Continued*)

**Table 1.** (Continued)

| | COVID-19 test negative or not tested | COVID-19 test positive |
|---|---|---|
| | **(n = 426,875)** | **(n = 1,324)** |
| ≥ 2 | 23,327 | 157 |
| | 5.5 % | 11.9 % |
| **Number of respiratory long-term conditions** | | |
| 0 | 373,026 | 1,120 |
| | 87.4 % | 84.6 % |
| 1 | 51,226 | 186 |
| | 12.0 % | 14.0 % |
| ≥ 2 | 2,623 | 18 |
| | 0.6 % | 1.4 % |
| **Number of medications** | | |
| 0 | 121,288 | 296 |
| | 28.5 % | 22.4 % |
| 1–3 | 197,296 | 526 |
| | 46.3 % | 39.8 % |
| 4–6 | 76,265 | 298 |
| | 17.9 % | 22.5 % |
| 7–9 | 22,779 | 130 |
| | 5.3 % | 9.8 % |
| ≥ 10 | 8,538 | 72 |
| | 2 % | 5.4 % |

This table uses participants with COVID-19 positive tests as positive group and all other participants as negative group. All chi squared tests p<0.01.

## Multimorbidity and COVID-19

In the fully adjusted model (Model 2), compared to those with no LTCs, participants with 1 LTC had no higher risk of having a positive test for COVID-19 (RR 1.12 (0.96–1.30) p = 0.15), but those with ≥2 LTCs had a 48% higher risk (RR 1.48 (1.28–1.71) p<0.01) (Table 2). Compared to those with no cardiometabolic LTCs, those with 1 cardiometabolic LTC had a 28% higher risk (RR 1.28 (1.12–1.46) p<0.01), and those with ≥2 cardiometabolic LTCs had a 77% higher risk (RR 1.77 (1.46–2.15) p<0.01) (Table 2). Participants with one respiratory LTC had no greater risk of a positive COVID-19 test compared to those with no respiratory LTCs (RR 1.14 (0.97–1.33) p = 0.12; Table 2). There was a higher risk of having a positive COVID-19 test in those with ≥2 respiratory LTCs (RR 1.78 (1.10–2.88) p = 0.02) compared to 0 respiratory LTCs but this did not meet our p-value threshold of 0.01. However, it is a similar higher risk observed (78%) as for participants with ≥2 cardiometabolic conditions (77%); the lack of statistical significance may be explained by the much lower number of participants in the group with ≥2 respiratory LTCs (N = 2,641) compared to ≥2 cardiometabolic conditions (N = 23,484).

## Polypharmacy and COVID-19

Compared to those taking no medications, in the fully adjusted model (Model 2) there was a clear dose relationship whereby the risk of a COVID-19 positive test rose steadily with polypharmacy level (Table 2): 4–6 medications (RR 1.41 (1.18–1.67) p<0.01); 7–9 medications (RR 1.86 (1.49–2.33) p<0.01); and ≥10 medications (RR 2.42 (1.82–3.21) p<0.01).

**Table 2. Relative risk of positive COVID-19 test by LTC groups (Poisson regression).**

| Measure of Multimorbidity (n) | Model 1 RR (95% CI) | P value | Model 2 RR (95% CI) | P value |
|---|---|---|---|---|
| **Total number of LTCs** | | | | |
| 0 (149,177) | 1 (ref) | - | 1 (ref) | - |
| 1 (140,348) | 1.18 (1.02–1.36) | 0.03 | 1.12 (0.96–1.30) | 0.15 |
| ≥2 (137,080) | 1.73 (1.51–1.99) | *** | 1.48 (1.28–1.71) | *** |
| **Number of cardiometabolic LTCs** | | | | |
| 0 (301,136) | 1 (ref) | - | 1 (ref) | - |
| 1 (103,579) | 1.41 (1.24–1.60) | *** | 1.28 (1.12–1.46) | *** |
| ≥2 (23,484) | 2.17 (1.82–2.60) | *** | 1.77 (1.46–2.15) | *** |
| **Number of respiratory LTCs** | | | | |
| 0 (374,146) | 1 (ref) | - | 1 (ref) | - |
| 1 (51,412) | 1.20 (1.03–1.41) | 0.02 | 1.14 (0.97–1.33) | 0.12 |
| ≥2 (2,641) | 2.09 (1.31–3.33) | *** | 1.78 (1.10–2.88) | 0.02 |
| **Number of medications** | | | | |
| 0 (121,584) | 1 (ref) | - | 1 (ref) | - |
| 1–3 (192,822) | 1.13 (0.98–1.31) | 0.10 | 1.07 (0.93–1.24) | 0.36 |
| 4–6 (76,563) | 1.58 (1.34–1.87) | *** | 1.41 (1.18–1.67) | *** |
| 7–9 (22,909) | 2.24 (1.81–2.77) | *** | 1.86 (1.49–2.33) | *** |
| ≥ 10 (8,610) | 3.09 (2.37–4.03) | *** | 2.42 (1.82–3.21) | *** |

Model 1: Adjusted for age, sex, Townsend score, ethnicity, and assessment centre location. Model 2: As model 1 and additionally adjusted for smoking status, alcohol intake frequency, BMI, and physical activity. RR = Relative risk; CI = confidence interval; n = number of participants; LTC = long-term condition; Cardiometabolic LTC = diabetes, coronary heart disease, atrial fibrillation, chronic heart failure, chronic kidney disease, hypertension, stroke/TIA or peripheral vascular disease; Respiratory LTC = asthma, chronic obstructive pulmonary disease, chronic bronchitis, emphysema, or bronchiectasis.

***p<0.01 Note: These results show the RR of a positive COVID-19 test versus a negative COVID-19 test or not tested (counterfactual group contains both participants who have a negative COVID-19 test result and participants who were not tested (n = 426,875)).

## Interactions between COVID-19 prognostic factors and multimorbidity

For all risk factors examined, there was a general trend of higher risk of a positive COVID-19 test with increasing LTCs and at-risk subgroups (Fig 1; Table 3). Further, for all prognostic factors, in participants with ≥2 LTCs there was a higher risk of a positive COVID-19 test for each risk factor subgroup when examining ethnicity, physical activity, BMI, systolic blood pressure, and eGFR. For some factors, having ≥2 LTCs was associated with an increased risk of COVID-19 infection only in the at-risk sub-group: males when examining sex; >65 years when examining age; and current/previous smoker when examining smoking status. When observing the effect of socioeconomic status, significantly higher risk of COVID-19 infection was apparent in the more deprived quintiles. There was no evidence of a multiplicative interaction between all risk factors modelled and number of LTCs. However, there appeared to be an additive effect whereby the combination of multimorbidity and each prognostic factor was associated with greater risk of COVID-19 infection. Of those with no LTCs, being ≥65 years old was associated with 34% lower risk of COVID-19 compared with those <65 years old.

Similar results overall were observed in the secondary analysis, where the above analyses were repeated in those who were tested for COVID-19 versus those who were not (S1–S3 Tables).

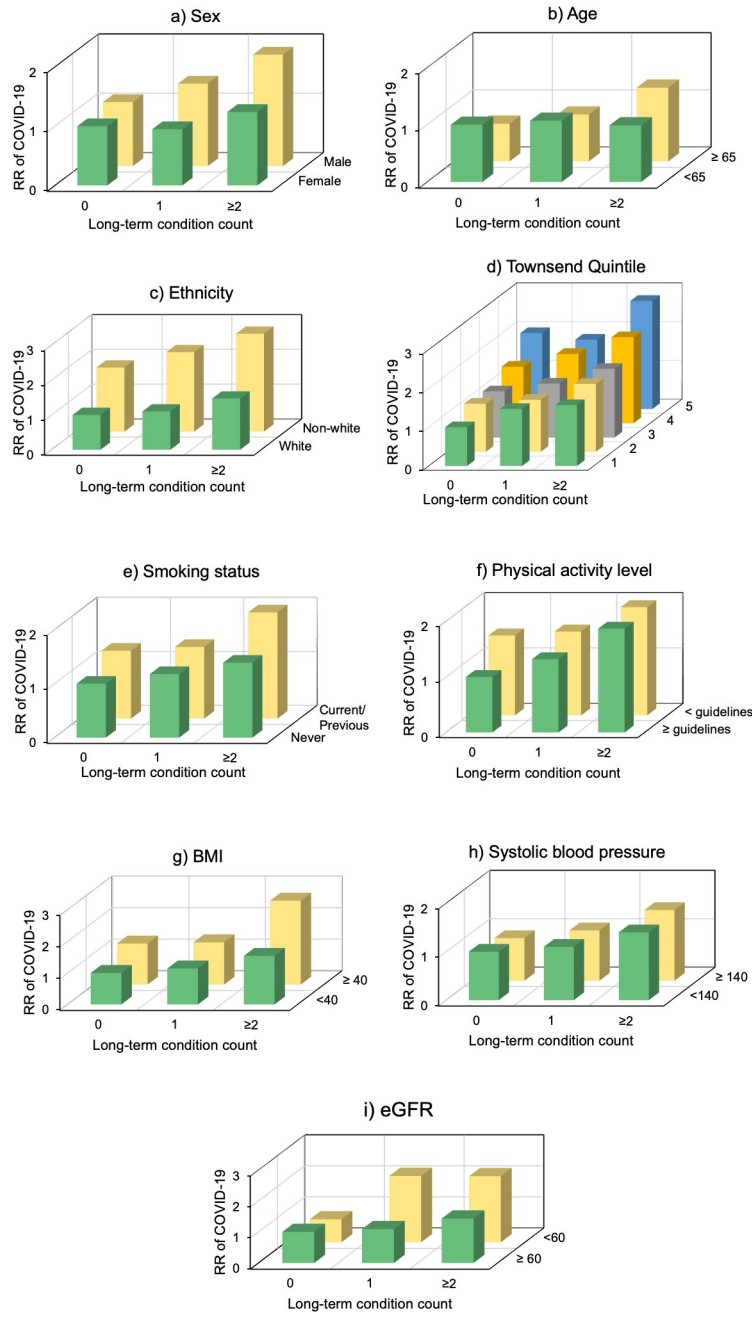

**Fig 1. Relative risk of positive COVID-19 test by long-term condition count and prognostic factors (Poisson regression).** Prognostic factors: a) Sex, b) Age (years), c) Ethnicity, d) Townsend quintile (1 least deprived; 5 most deprived), e) Smoking status, f) Physical activity level (based on UK guidelines), g) Body-mass index (BMI; $kg/m^2$), h) Systolic blood pressure (mmHg), and i) estimated glomerular filtration rate (eGFR; $ml/min/1.73m^2$). Models were adjusted for sex, age, ethnicity, Townsend score, smoking status, alcohol intake frequency, physical activity, BMI, and assessment centre location.

**Table 3. Relative risk of positive COVID-19 test by LTC count and prognostic factors (Poisson regression).**

| Prognostic factor | LTC count | Prognostic factor subgroup | N | Relative risk (95% CI) | P value |
|---|---|---|---|---|---|
| **Sex** | 0 | Female | 80,212 | 1 (ref) | |
| | | Male | 68,965 | 1.08 (0.87–1.33) | 0.51 |
| | 1 | Female | 76,383 | 0.95 (0.77–1.18) | 0.67 |
| | | Male | 63,965 | 1.39 (1.13–1.71) | *** |
| | ≥2 | Female | 77,492 | 1.24 (1.01–1.51) | 0.04 |
| | | Male | 59,588 | 1.88 (1.54–2.29) | *** |
| **Age at COVID-19 test (years)** | 0 | < 65 | 16,470 | 1 (ref) | |
| | | ≥ 65 | 132,707 | 0.66 (0.53–0.82) | *** |
| | 1 | < 65 | 25,029 | 1.07 (0.87–1.31) | 0.51 |
| | | ≥ 65 | 115,319 | 0.82 (0.68–1.00) | 0.05 |
| | ≥2 | < 65 | 36,865 | 0.99 (0.79–1.25) | 0.96 |
| | | ≥ 65 | 100,215 | 1.29 (1.09–1.53) | *** |
| **Ethnicity** | 0 | White | 138,677 | 1 (ref) | |
| | | Other | 9,492 | 1.84 (1.34–2.54) | *** |
| | 1 | White | 131,644 | 1.10 (0.94–1.29) | 0.24 |
| | | Other | 8,055 | 2.28 (1.67–3.11) | *** |
| | ≥2 | White | 129,033 | 1.47 (1.26–1.72) | *** |
| | | Other | 7,408 | 2.81 (2.09–3.78) | *** |
| **Townsend quintile (1-least deprived; 5-most deprived)** | 0 | 1 | 30,995 | 1 (ref) | |
| | | 2 | 31,206 | 1.24 (0.84–1.82) | 0.28 |
| | | 3 | 30,394 | 1.19 (0.80–1.75) | 0.39 |
| | | 4 | 29,633 | 1.46 (1.01–2.12) | 0.04 |
| | | 5 | 26,956 | 1.96 (1.37–2.80) | *** |
| | 1 | 1 | 28,723 | 1.47 (1.00–2.14) | 0.05 |
| | | 2 | 28,898 | 1.34 (0.92–1.97) | 0.13 |
| | | 3 | 28,605 | 1.39 (0.95–2.04) | 0.09 |
| | | 4 | 27,950 | 1.79 (1.25–2.46) | *** |
| | | 5 | 26,005 | 1.79 (1.19–2.70) | *** |
| | ≥2 | 1 | 25,310 | 1.57 (1.07–2.30) | 0.02 |
| | | 2 | 26,585 | 1.75 (1.21–2.52) | *** |
| | | 3 | 26,771 | 1.77 (1.23–2.55) | *** |
| | | 4 | 27,764 | 2.22 (1.57–3.15) | *** |
| | | 5 | 20,670 | 2.79 (2.00–3.90) | *** |
| **Smoking status** | 0 | never | 88,737 | 1 (ref) | |
| | | current/previous | 59,500 | 1.26 (1.02–1.57) | 0.03 |
| | 1 | never | 77,915 | 1.18 (0.96–1.44) | 0.11 |
| | | current/previous | 61,781 | 1.33 (1.08–1.65) | 0.01 |
| | ≥2 | never | 68,324 | 1.39 (1.14–1.70) | *** |
| | | current/previous | 67,956 | 1.97 (1.62–2.38) | *** |
| **Physical activity level** | 0 | ≥ guidelines | 80,715 | 1 (ref) | |
| | | < guidelines | 36,507 | 1.44 (1.09–1.91) | 0.01 |
| | 1 | ≥ guidelines | 74,324 | 1.32 (0.94–1.83) | 0.11 |
| | | < guidelines | 32,998 | 1.51 (1.13–2.00) | *** |
| | ≥2 | ≥ guidelines | 68,226 | 1.87 (1.40–2.67) | *** |
| | | < guidelines | 29,622 | 1.95 (1.43–2.67) | *** |

*(Continued)*

**Table 3.** (Continued)

| Prognostic factor | LTC count | Prognostic factor subgroup | N | Relative risk (95% CI) | P value |
|---|---|---|---|---|---|
| BMI (kg/m$^2$) | 0 | <40 | 146,986 | 1 (ref) | |
| | | ≥40 | 1,094 | 1.30 (0.49–3.50) | 0.60 |
| | 1 | <40 | 137,865 | 1.14 (0.99–1.33) | 0.08 |
| | | ≥40 | 1,912 | 1.34 (0.63–2.85) | 0.44 |
| | ≥2 | <40 | 131,416 | 1.54 (1.33–1.78) | *** |
| | | ≥40 | 4,865 | 2.66 (1.88–3.76) | *** |
| Systolic blood pressure (mm Hg) | 0 | <140 | 97,526 | 1 (ref) | |
| | | ≥140 | 47,162 | 0.88 (0.69–1.11) | 0.28 |
| | 1 | <140 | 78,786 | 1.10 (0.91–1.32) | 0.33 |
| | | ≥140 | 57,921 | 1.04 (0.84–1.29) | 0.73 |
| | ≥2 | <140 | 68,956 | 1.40 (1.16–1.69) | *** |
| | | ≥140 | 63,979 | 1.46 (1.32–1.78) | *** |
| eGFR (ml/min/1.73m$^2$) | 0 | ≥ 60 | 137,083 | 1 (ref) | |
| | | < 60 | 1,270 | 0.74 (0.18–2.96) | 0.67 |
| | 1 | ≥ 60 | 128,860 | 1.09 (0.93–1.28) | 0.26 |
| | | < 60 | 2,108 | 2.14 (1.17–3.92) | 0.01 |
| | ≥2 | ≥ 60 | 123,182 | 1.43 (1.22–1.67) | *** |
| | | < 60 | 4,981 | 2.13 (1.46–3.12) | *** |

Models were adjusted for sex, age, ethnicity, Townsend score, smoking status, alcohol intake frequency, physical activity, BMI, and assessment centre location.

LTC = long-term condition; BMI = body mass index; eGFR = estimated glomerular filtration rate; guidelines = UK guidelines of 150 min/week moderate or 75 min/week vigorous physical activity.

***p<0.01.

## Discussion

### Summary of key findings

This study showed that participants with multimorbidity (≥2 LTCs) had a 48% higher risk of a positive COVID-19 test, those with cardiometabolic multimorbidity had a 77% higher risk, and those with respiratory multimorbidity a 78% higher risk (albeit not statistically significant, p = 0.02), compared to those without that type of multimorbidity. Importantly, those from non-white ethnicities with multimorbidity had nearly three times the risk of having COVID-19 infection compared to those of white ethnicity, suggesting that those from minority ethnic groups with multimorbidity are at particular risk. COVID-19 prognostic factors appeared to have an additive effect by further increasing the risk of a positive COVID-19 test in those with multimorbidity.

### Comparison with previous literature

Previous literature has suggested that the presence of single LTCs such as hypertension, diabetes, or COPD increase the risk of COVID-19 [33]. In an English primary care research cohort of 3,802 patients, confirmed COVID-19 infections were associated with male sex, black ethnicity, deprivation, chronic kidney disease, and urban setting [30]. A preprint report of linked primary care data on 17,425,445 adults in England, showed that older age, male sex, deprivation, and black and Asian ethnicity were associated with higher risk of COVID-19 related in-hospital deaths [11]. However, our study is the first to show a higher risk of a positive COVID-19 test in those with ≥2 LTCs and particularly in those with cardiometabolic multimorbidity.

It has also been suggested that the presence of multimorbidity could further increase the risk of adverse outcomes for people with COVID-19 [33]. Guan et al, in a cohort of 1,590 hospital patients with confirmed COVID-19, found a higher risk of a composite COVID-19 outcome (intensive care admission, invasive ventilation, or death) in those with 1 (HR 1.79 (95%CI 1.16–2.77)) and 2 (HR 2.59 (95%CI 1.61–4.17) comorbidities [9]. However, while they provided details of specific LTCs most likely to be present, especially in severe cases of infection (hypertension, cardiovascular/cerebrovascular diseases, diabetes, hepatitis B infections, COPD, chronic kidney diseases and malignancy), they did not describe which patterns of multimorbidity were associated with the greatest risk. A report on 3,200 patients with COVID-19 from Italy showed that, of the 481 patients who died, 48.6% had 3 or more comorbidities. However, again, while it listed the specific LTCs associated with increased risk, such as ischaemic heart disease and diabetes, it did not describe which patterns of multimorbidity were associated with the highest risk [13].

## Strengths and limitations

As a large prospective cohort with rich demographic, lifestyle, health, and anthropometric data linked to COVID-19 test results, UK Biobank provides a valuable opportunity to examine the predictors for COVID-19 [34, 35]. In particular, the rich data allowed examination of the risk of a positive COVID-19 test in those with multimorbidity and the influence of a range of known sociodemographic, lifestyle and physiological risk factors for COVID-19. However, our study has a number of limitations. Firstly, the proportion of UK Biobank participants with COVID-19 test data is currently low (1.05%) which resulted in wide confidence intervals for groups with few participants. Secondly, the denominator for the test group included all those who had a negative test result as well as those who were not tested at all. At the time for which COVID-19 test data are available, the strategy in the UK had been to only test those in hospital (emergency department and inpatient) settings. This means the positive COVID-19 participants are likely to have had sufficiently severe clinical signs and symptoms to justify hospital assessment and those with COVID-19 but with mild symptoms are less likely to have been tested. Our results are therefore likely to reflect the associations with more severe COVID-19 disease. It is not known if the associations identified in this study would be similar for those with milder COVID-19 disease. Thirdly, exposures and moderators examined here were assessed at baseline only and may have changed during follow up. Depending on the direction and level of change since recruitment, more up-to-date data on exposures (e.g. smoking, alcohol intake, physical activity, and post code of residence) could have provided different results. However, LTC and medication count are likely to have remained the same or increased with time and therefore our effect size estimates may be conservative. Fourthly, UK Biobank participants are not representative of the general population and are acknowledged to be mostly white British, more affluent, and healthier than the general population [36]. Consequently, absolute values may not be generalisable, however, effect size estimates will be and strongly agree with more representative cohorts [37]. Finally, participants in this study were aged between 48–86 years old and associations between multimorbidity and COVID-19 may be different for younger age groups. This may be particularly important for participants from more deprived backgrounds who are more likely to have multimorbidity at a younger age [38, 39].

## Implications

Our work has clinical and practical implications as more countries navigate lifting COVID-19 restrictions. It demonstrates that multimorbidity, particularly cardiometabolic multimorbidity and polypharmacy, are strongly associated with COVID-19 infection and suggests that those who have multimorbidity coupled with additional risk factors, such as non-white ethnicity, are

at increased risk. Such individuals should be particularly stringent in adhering to preventive measures, such as physical distancing and hand hygiene. Our findings also have implications for clinicians, occupational health and employers when considering work-place environments, appropriate advice for patients, and adaptations that might be required to protect such staff.

Future research is needed to corroborate these findings in other countries and in people from different ethnic backgrounds. We know that patterns of multimorbidity differ across ethnic groups and have different associations with mortality, so this merits further investigation [40]. We also need to explore the implications of different patterns of multimorbidity on COVID-19 related health care outcomes in the short and long term.

## Conclusion

This study suggests that multimorbidity, cardiometabolic disease, and polypharmacy are associated with COVID-19. Those with multimorbidity who were also of non-white ethnicity, from the most socioeconomically deprived backgrounds, those who were severely obese, or who had reduced renal function, had more than twice the risk of COVID-19 infection.

More work is required to develop risk stratification for COVID-19 in people with different patterns of multimorbidity in order to better define those individuals who would benefit from enhanced preventive measures in public, work, and residential spaces.

## Supporting information

**S1 Table. Cohort characteristics by COVID-19 testing.**
(DOCX)

**S2 Table. Relative risk of COVID-19 testing by multimorbidity (Poisson regression).**
(DOCX)

**S3 Table. Relative risk of COVID-19 testing by LTC category and prognostic factors.**
(DOCX)

## Acknowledgments

The authors would like to thank the participants of UK Biobank and the UK Biobank for access to this data.

## Author Contributions

**Conceptualization:** Ross McQueenie, Hamish M. E. Foster, Bhautesh D. Jani, Srinivasa Vittal Katikireddi, Naveed Sattar, Jill P. Pell, Catherine A. O'Donnell, Frances S. Mair, Barbara I. Nicholl.

**Formal analysis:** Ross McQueenie.

**Methodology:** Ross McQueenie, Bhautesh D. Jani, Naveed Sattar, Jill P. Pell, Frederick K. Ho, Claire L. Niedzwiedz, Claire E. Hastie, Jana Anderson, Patrick B. Mark, Michael Sullivan, Catherine A. O'Donnell, Frances S. Mair, Barbara I. Nicholl.

**Project administration:** Hamish M. E. Foster, Barbara I. Nicholl.

**Supervision:** Bhautesh D. Jani, Catherine A. O'Donnell, Frances S. Mair, Barbara I. Nicholl.

**Validation:** Hamish M. E. Foster, Barbara I. Nicholl.

**Visualization:** Ross McQueenie, Hamish M. E. Foster, Bhautesh D. Jani, Frederick K. Ho, Catherine A. O'Donnell, Frances S. Mair, Barbara I. Nicholl.

**Writing – original draft:** Ross McQueenie, Hamish M. E. Foster, Catherine A. O'Donnell, Frances S. Mair, Barbara I. Nicholl.

**Writing – review & editing:** Ross McQueenie, Hamish M. E. Foster, Bhautesh D. Jani, Srinivasa Vittal Katikireddi, Naveed Sattar, Jill P. Pell, Frederick K. Ho, Claire L. Niedzwiedz, Claire E. Hastie, Jana Anderson, Patrick B. Mark, Michael Sullivan, Catherine A. O'Donnell, Frances S. Mair, Barbara I. Nicholl.

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
