## [Decision Letter · Decision Letter 0]

8 Jul 2020

PONE-D-20-17811

­­Multimorbidity, Polypharmacy, and COVID-19 infection within the UK Biobank cohort.

PLOS ONE

Dear Dr. Foster,

Thank you for submitting your manuscript to PLOS ONE. After careful consideration, we feel that it has merit but does not fully meet PLOS ONE’s publication criteria as it currently stands. Therefore, we invite you to submit a revised version of the manuscript that addresses the points raised during the review process.

We look forward to receiving your revised manuscript.

Kind regards,

Ying-Mei Feng

Academic Editor

PLOS ONE

Journal Requirements:

"I have read the journal's policy and one author (JPP) of this manuscript has the following competing interest: JPP is a member of the Scientific Advisory Committee of UK Biobank. All other authors have no potential, perceived, or real conflicts of interest."

Reviewers' comments:

Reviewer's Responses to Questions

**Comments to the Author**

1. Is the manuscript technically sound, and do the data support the conclusions?

Reviewer #1: Yes

2. Has the statistical analysis been performed appropriately and rigorously? 

Reviewer #1: I Don't Know

3. Have the authors made all data underlying the findings in their manuscript fully available?

Reviewer #1: Yes

4. Is the manuscript presented in an intelligible fashion and written in standard English?

Reviewer #1: Yes

5. Review Comments to the Author

Reviewer #1: This paper reports an analysis undertaken using the UK Biobank cohort to examine the relationship between multiple long-term conditions (multimorbidity) and the likelihood of testing positive to COVID-19. The aim of the paper is to contribute to the evidence base in an effort to assist in developing more robust risk stratification models to help protect those most vulnerable to COVID-19.

This paper reports that multimorbidity esp when combined with socioeconomic deprivation, renal impairment and BMI of 40 or more (measured a decade or more earlier) is associated with a greater likelihood of being tested and testing positive for COVID-19. Whilst they are not unexpected findings, they make a useful contribution to the literature as we build the evidence base to support clinical decision making during the COVID-19 pandemic.

The paper could be improved by addressing these questions:

• Baseline data are reported as being collected between 2006-2010 – am I correct in thinking that the LTC status was established then? That is 10 years or more ago. If that is the case, people may have additional LTCs that you are not aware of. How have you accounted for this? The same questions should be answered for medications, physical activity, BMI, BP etc. There is one mention of this limitation in the discussion. This needs to be expanded upon and made clearer from the outset when describing the study.

• Can you explain how you decided the category cut-offs (e.g. age of 65yo?)

• The outcome is COVID-19 test or COVID-19 positive – is there information about whether the person was unwell or not? Can you link with death or hospitalisation data or visits to GPs? This would strengthen the paper.

• Being limited to the age group of 48-86y is a significant limitation and should be discussed more fully. A key clinical question is whether and to what extent MM increases risk of COVID-19 for younger people (esp those from deprived areas), this is worth commenting on. I also found it very interesting to note that you report that if you are older than 65 and have no LTC then you are less likely (RR=0.66) to have a positive COVID-19 test – this is worthy of discussion too and what it might mean for individuals and clinicians.

• Why did you exclude the 5 patients from Scotland and Wales?

• The group who were not tested is likely to include people who did have COVID-19 but were not ill enough to qualify for testing. This should be discussed in more depth in the discussion.

6. PLOS authors have the option to publish the peer review history of their article (what does this mean?). If published, this will include your full peer review and any attached files.

Reviewer #1: No

---

## [Author Response · Author response to Decision Letter 0]

15 Jul 2020

Manuscript Titled: Multimorbidity, Polypharmacy, and COVID-19 infection within the UK Biobank Cohort.

We have pleasure in resubmitting our revised manuscript and our response to reviewers’ comments. In addition to our point by point response to the reviewers’ comments, we can also confirm that the following journal requirements have been met in the revised manuscript: 

1. The revised manuscript is in the required PLOS One style. 

2. Updated Conflicts of interest statement:

“JPP is a member of the Scientific Advisory Committee of UK Biobank. This does not alter our adherence to PLOS ONE policies on sharing data and materials. All other authors have no potential, perceived, or real conflicts of interest.”

3. Updated Data availability statement: 

“The data in this study is owned by the UK Biobank (www.ukbiobank.ac.uk) and as researchers we are not entitled to republish or otherwise make available any UK Biobank data at the individual participant level (https://www.ukbiobank.ac.uk/wp-content/uploads/2013/10/ukbiobank-data-management.pdf). However, any bona fide researcher can apply to use the UK Biobank resource for health-related research that is in the public interest by following the registration and access link https://bbams.ndph.ox.ac.uk/ams/.”

4. Captions for our Supporting Information files have been included at the end of the revised manuscript, and in-text citations match these captions. 

Thank you for your consideration and we look forward to your decision in due course. 

Response to reviewers

Reviewer comment:

Reviewer #1: This paper reports an analysis undertaken using the UK Biobank cohort to examine the relationship between multiple long-term conditions (multimorbidity) and the likelihood of testing positive to COVID-19. The aim of the paper is to contribute to the evidence base in an effort to assist in developing more robust risk stratification models to help protect those most vulnerable to COVID-19.

This paper reports that multimorbidity esp when combined with socioeconomic deprivation, renal impairment and BMI of 40 or more (measured a decade or more earlier) is associated with a greater likelihood of being tested and testing positive for COVID-19. Whilst they are not unexpected findings, they make a useful contribution to the literature as we build the evidence base to support clinical decision making during the COVID-19 pandemic.

Authors response:

We thank the reviewer for their comment.

Reviewer comment:

The paper could be improved by addressing these questions:

• Baseline data are reported as being collected between 2006-2010 – am I correct in thinking that the LTC status was established then? That is 10 years or more ago. If that is the case, people may have additional LTCs that you are not aware of. How have you accounted for this? The same questions should be answered for medications, physical activity, BMI, BP etc. There is one mention of this limitation in the discussion. This needs to be expanded upon and made clearer from the outset when describing the study.

Authors’ response:

We thank the reviewer for pointing out this issue. It is correct that baseline data including LTC counts were collected between 2006-2010. We agree that the LTC counts here will represent underestimates as LTCs tend to accumulate with age. We are unable to account for this other than to acknowledge this as a limitation of this dataset. However, LTC counts presented are underestimates and the associations presented are therefore also likely to be underestimates due to misclassification and regression dilution bias. This is described in the limitations section (page 20, line 390): ‘However, LTC and medication count are likely to have remained the same or increased with time and therefore our effect size estimates may be conservative.’

We describe the measurement of LTCs (page 7, line 155) and polypharmacy (page 8, line 165) as being at baseline in the methods. We have made this clearer by adding a further sentence (page 8, line 169) to state this fact again:

‘As for LTC and polypharmacy measures, all exposures were based on assessment at the time of recruitment.’

We have added a further sentence in the strengths and limitations section to expand on this limitation (page 20, line 387): 

‘Depending on the direction and level of change since recruitment, more up-to-date data on exposures (e.g. smoking, alcohol intake, physical activity, and post code of residence) could have provided different results.’

We hope that this limitation is now clearer for readers who are interpreting our results.

Reviewer comment:

• Can you explain how you decided the category cut-offs (e.g. age of 65yo?)

Authors’ response:

We decided to use the age cut-off used by the Centre for Disease Control: >65 years representing higher risk. (Centers for Disease Control and Prevention: Coronavirus Disease 2019 (reference number 14 https://www.cdc.gov/coronavirus/2019-ncov/need-extra-precautions/older-adults.html)

Reviewer comment:

• The outcome is COVID-19 test or COVID-19 positive – is there information about whether the person was unwell or not? Can you link with death or hospitalisation data or visits to GPs? This would strengthen the paper.

Authors’ response:

We thank the reviewer for this comment and agree that a more direct measure of severity like death or hospitalisation would strengthen the paper. Unfortunately, death and hospitalisation data were not available. However, a COVID-19 test during the study period here was ipso facto a proxy for more severe disease due to the testing strategy in place at the time as only those in hospital were being tested for COVID-19. We direct the reviewer to our comment on this issue in our strengths and limitations section (page 20, line 378): ‘Secondly, the denominator for the test group included all those who had a negative test result as well as those who were not tested at all. At the time for which COVID-19 test data are available, the strategy in the UK had been to only test those in hospital (emergency department and inpatient) settings. This means the positive COVID-19 participants are likely to have had sufficiently severe clinical signs and symptoms to justify hospital assessment and those with COVID-19 but with mild symptoms are less likely to have been tested. Our results are therefore likely to reflect the associations with more severe COVID-19 disease.’

In addition, the novelty of our work is that it is the first to show a higher risk of COVID-19 infection in those with multimorbidity. There is already another preprint looking at multimorbidity and mortality (reference number 11).

Reviewer comment:

• Being limited to the age group of 48-86y is a significant limitation and should be discussed more fully. A key clinical question is whether and to what extent MM increases risk of COVID-19 for younger people (esp those from deprived areas), this is worth commenting on. I also found it very interesting to note that you report that if you are older than 65 and have no LTC then you are less likely (RR=0.66) to have a positive COVID-19 test – this is worthy of discussion too and what it might mean for individuals and clinicians.

Authors’ response:

We thank the reviewer for this comment. We agree it would strengthen the paper if a wider age range were available. And we agree that this is likely to be more important for more deprived populations who have higher mortality and multimorbidity at younger ages. 

We have added the following to the strengths and limitations section (page 21, line 397):

‘Finally, participants in this study were aged between 48-86 years old and associations between multimorbidity and COVID-19 may be different for younger age groups. This may be particularly important for participants from more deprived backgrounds who are more likely to have multimorbidity at a younger age.(38,39)’

We thank the reviewer for highlighting the result of those ≥65 years old having a lower risk of COVID-19. We have added the following to the results section…

Page 16, line 296: ‘Of those with no LTCs, being ≥65 years old was associated with 34% lower risk of COVID-19 compared with those <65 years old.’

Reviewer comment:

• Why did you exclude the 5 patients from Scotland and Wales?

Authors’ response:

We only analysed those participants who were recruited from England as test data were not available for participants currently living in Scotland and Wales. The 5 participants with test data who were recruited from Scotland and Wales must have moved or travelled to England at some point in order to have COVID-19 test data available here. We adjusted for assessment centre location in our models in an attempt to reduce confounding due to geographical location and local epidemics. In order to simplify analyses, we excluded the few who were recruited outside England but had COVID-19 data.

Reviewer comment:

• The group who were not tested is likely to include people who did have COVID-19 but were not ill enough to qualify for testing. This should be discussed in more depth in the discussion.

Authors’ response:

Please see our response to the comment above regarding lack of outcome data that measures disease severity more directly. 

We have also added the following to the strengths and limitations to expand on this issue more (page 20, line 381):

‘It is not known if the associations identified in this study would be similar for those with milder COVID-19 disease.’

---

## [Decision Letter · Decision Letter 1]

11 Aug 2020

­­Multimorbidity, Polypharmacy, and COVID-19 infection within the UK Biobank cohort.

PONE-D-20-17811R1

Dear Dr. Foster,

We’re pleased to inform you that your manuscript has been judged scientifically suitable for publication and will be formally accepted for publication once it meets all outstanding technical requirements.

Kind regards,

Ying-Mei Feng

Academic Editor

PLOS ONE

---

## [Editor Report · Acceptance letter]

12 Aug 2020

PONE-D-20-17811R1 

­­­­Multimorbidity, polypharmacy, and COVID-19 infection within the UK Biobank cohort 

Dear Dr. Foster:

I'm pleased to inform you that your manuscript has been deemed suitable for publication in PLOS ONE. Congratulations! Your manuscript is now with our production department. 

Kind regards, 

on behalf of

Dr Ying-Mei Feng 

Academic Editor

PLOS ONE